# Wnt Signaling Rescues Amyloid Beta-Induced Gut Stem Cell Loss

**DOI:** 10.3390/cells11020281

**Published:** 2022-01-14

**Authors:** Prameet Kaur, Ellora Hui Zhen Chua, Wen Kin Lim, Jiarui Liu, Nathan Harmston, Nicholas S. Tolwinski

**Affiliations:** 1Division of Science, Yale-NUS College, Singapore 138527, Singapore; prameet.k87@gmail.com (P.K.); ellora.chua@yale-nus.edu.sg (E.H.Z.C.); wenkinlim@gmail.com (W.K.L.); jiarui_liu@u.yale-nus.edu.sg (J.L.); nathan.harmston@yale-nus.edu.sg (N.H.); 2Program in Cancer and Stem Cell Biology, Duke-NUS Medical School, Singapore 169857, Singapore

**Keywords:** Alzheimer’s disease, amyloid-β, Wnt, *Drosophila*

## Abstract

Patients with Alzheimer’s disease suffer from a decrease in brain mass and a prevalence of amyloid-β plaques. These plaques are thought to play a role in disease progression, but their exact role is not entirely established. We developed an optogenetic model to induce amyloid-β intracellular oligomerization to model distinct disease etiologies. Here, we examine the effect of Wnt signaling on amyloid in an optogenetic, *Drosophila* gut stem cell model. We observe that Wnt activation rescues the detrimental effects of amyloid expression and oligomerization. We analyze the gene expression changes downstream of Wnt that contribute to this rescue and find changes in aging related genes, protein misfolding, metabolism, and inflammation. We propose that Wnt expression reduces inflammation through repression of Toll activating factors. We confirm that chronic Toll activation reduces lifespan, but a decrease in the upstream activator Persephone extends it. We propose that the protective effect observed for lithium treatment functions, at least in part, through Wnt activation and the inhibition of inflammation.

## 1. Introduction

Alzheimer’s disease (AD) is an age-related disease affecting millions of people worldwide [1,2]. No effective therapy exists, despite some recent clinical trials and one controversial new drug which has been recently approved [3,4,5]. AD drug development has primarily been based on the amyloid cascade hypothesis, with many drugs targeting amyloid beta (Aβ) directly [6,7]. The hypothesis postulates that the extracellular deposition of Aβ leads to neuronal cell death driving AD [8], but whether these plaques are causative remains controversial, due to observations of plaques in asymptomatic individuals [9].

Aβ extracellular aggregates form macroscopic plaques that correlate with cell loss, but Aβ can also form smaller oligomers with soluble oligomers showing the highest toxicity [10]. We recently described an optogenetic method to study Aβ protein oligomerization in vivo in several model organisms [11]. Optogenetics refers to genes that are modified with a light responsive protein domain, allowing the spatial and temporal regulation of proteins [12]. Temporal and spatial control is achieved by light exposure of a specific wavelength, without the need to introduce other external agents [13,14]. We used a modified version of the *Arabidopsis thaliana* cryptochrome 2 (CRY2) protein, which clusters in response to blue light [15].

The Wnt signaling pathway defines several developmental processes in a variety of organisms through two or more branches [16,17]. The canonical pathway defines cell fates, and the non-canonical pathway is involved in cell polarity; for example, in *Drosophila* intestinal stem cells, two Wnt pathway branches are required for stem cell maintenance and regeneration [18,19,20,21]. In adults, Wnts play a role in a variety of diseases in addition to cancer [16,22,23]. Alzheimer’s disease and Wnt were linked [24], where Wnt regulates a number of stem cell processes, lipid, glucose, and brain signaling pathways [17,25,26,27] through a complex signaling cascade [28,29,30]. In AD, GSK3 and TCF, the regulatory kinase and the main Wnt transcription factor respectively, were linked with disease risk [31,32]. These findings suggested that Wnt modulation could affect the disease state [33,34].

Our previous studies focused on neurogenesis and metabolism, as well as interventions that could ameliorate the condition [11,35]. We showed that lithium worked well to extend the lifespan of *C. elegans* and *Drosophila* in optogenetic Aβ models. We posited that the Wnt signaling pathway may be involved but did not delineate the pathway through which lithium worked. Here, we present evidence that Wnt signaling can ameliorate the detrimental effects of Aβ oligomerization by promoting stem cell homeostasis and preventing inflammation. We speculate that Wnt activators could function in various tissues affected by amyloid or amyloidosis.

## 2. Materials and Methods

### 2.1. Crosses and Expression of UAS Construct

Optogenetic transgenes were generated as previously described in [11,36]. Expression was driven by Elav-GAL4, the neuronal driver, *Escargot*-GAL4 [37], and tubP-GAL80^ts^ [38]. All additional stocks were obtained from the Bloomington *Drosophila* Stock Center (NIH P40OD018537) that were used in this study.

Fly lines:
Elav-Gal4: BDSC 458 [39]Repo-QF2: BDSC 66477 [40]QUAS-6XGFP: BDSC 52263 [41]UAS-Aβ-CRY2-mCherry: [11]UAS-mCD8.RFP: BDSC 32220 [42]UAS-wg: BDSC 5918 [43]UAS-Td-Tomato: BDSC 36328 (Joost Schulte and Katharine Sepp)Tl(CRISPaint.T2A-GAL4)wg: BDSC #83627 [44]UAS-10XGFP: BDSC 32185 [42]Esg-Gal4, UAS-GFP; tub-Gal80^ts^, UAS-dCas9.VPR: BDSC 67069 [45]UAS-myr::tdTomato: BDSC 32222 [42]UAS-Toll-Cry2-mCherry: [46]Arm-Gal4: BDSC 1560 [47]ArmGal4; tub-GAL80^ts^: BDSC 86327 [38]TRiP.HMC03615 attP40 Persephone RNAi [48]TRiP.HM05191 attP2 Dif RNAi [48]UAS-Dsh [49].

Fly crosses performed were:(1)Elav-Gal4; Repo-QF2, QUAS-GFP × UAS-Aβ-CRY2-mCh(2)Elav-Gal4, UAS-RFP × UAS-wg(3)Elav-Gal4 × UAS-Td-Tomato(4)Tl(CRISPaint.T2A-GAL4)wg × UAS-10XGFP(5)w; UAS-TdTomato X esg-Gal4, UAS-GFP; tubP-GAL80^ts^(6)w; UAS-Aβ^1−42^-CRY2-mCh X esg-Gal4, UAS-GFP; tubP-GAL80^ts^(7)w; UAS-wg; UAS-Aβ^1−42^-CRY2-mCh X esg-Gal4, UAS-GFP; tub-GAL80^ts^(8)w; UAS-wg, UAS-myr-Tomato × esg-Gal4, UAS-GFP; tub-GAL80^ts^(9)w; UAS-wg × esg-Gal4, UAS-GFP; tub-GAL80^ts^(10)w; UAS-Toll-Cry2-mCh × esg-Gal4, UAS-GFP; tub-GAL80^ts^(11)w; UAS-Toll-Cry2-mCh × ArmGal4; tub-GAL80^ts^(12)w; UAS-Td-Tomato × esg-Gal4, UAS-GFP; tub-GAL80^ts^(13)w; UAS-Td-Tomato × ArmGal4; tub-GAL80^ts^(14)w; UAS-Persephone-RNAi × ArmGal4; tub-GAL80^ts^(15)w; UAS-Dif-RNAi × ArmGal4; tub-GAL80^ts^.

### 2.2. Light-Sheet Microscopy

Embryos at stage 11 were selected using halocarbon oil (Sigma), dechorionated and mounted into the Lightsheet Z.1 (Carl Zeiss, Oberkochen, Germany) microscope, and imaged with a 40× W Plan-Apochromat 40× 1.0 UV–VIS detection objective [30,50,51]. Image data were processed using the maximum intensity projection function of ZEN 2014 SP software (Carl Zeiss, Oberkochen, Germany).

#### 2.2.1. Gut Preparations and Fluorescence Microscopy

Adult fly midguts from at least 3 flies were dissected and imaged at the 25th percentile from the anterior midgut on the Zeiss LSM800 (Carl Zeiss, Oberkochen, Germany), as described by Bunnag et al. 2020. Images were processed using the ZEN 2014 SP1 software (Carl Zeiss, Oberkochen, Germany). Image quantification was done in ImageJ (Developed by the National Institutes of Health, Bethesda, MD, USA [52]).

#### 2.2.2. RNA Preparation and RNA-Sequencing

Flies were dissected in 1 × PBS. RNA was isolated from at least 10 midguts using the Isolate II RNA Mini kit (Bioline, London, UK). The extracted RNA was quantified using Nanodrop (Thermo Fisher Scientific Waltham, Massachusetts, MA, USA). Library preparation was performed using 1 µg of total RNA, and sequencing was performed using an Illumina HiSeq 4000 System (2 × 151 bp read length, 40 million reads per sample) by NovogeneAIT Genomics (Singapore).

#### 2.2.3. RNA-Seq Analysis

Data processing and QC: RNA-seq was aligned against BDGP6.22 (Ensembl version 97) using STAR v2.7.1a [53], and quantified using RSEM v1.3.1 [54]. Reads mapping to genes annotated as rRNA, snoRNA, or snRNA were removed. Genes which had less than 10 reads mapping on average across all samples were also removed. A differential expression analysis was performed using DESeq2 [55]. The likelihood ratio test (LRT) was used to identify any genes that show change in expression across the different conditions. Pairwise comparisons were performed using a Wald test, with independent filtering. To control for false positives due to multiple comparisons in the genome-wide differential expression analysis, the false discovery rate (FDR) was computed using the Benjamini–Hochberg procedure. The gene level counts were transformed using a regularized log transformation, converted to z-scores, and clustered using partitioning around medoids (PAM), using correlation distance as the distance metric.

Functional enrichment analysis: Gene ontology (GO) and KEGG pathway enrichments for each cluster were performed using EnrichR [56,57,58]. Terms with an FDR < 10% were defined as significantly enriched.

#### 2.2.4. Lifespan Studies

*Drosophila* were counted daily for the number of dead subjects and the number of censored subjects (excluded from the study). *Drosophila* that failed to respond to taps were scored as dead, and those stuck to the food were censored. Lifespan analysis was performed using OASIS 2 (Online Application for Survival Analysis 2) [59]. Raw numbers are available for Appendix A.

## 3. Results

To study the role of intracellular Aβ, we developed a system of expression coupled with light inducible oligomerization (Figure 1a). Previously, we showed a distinction between the simple overexpression of Aβ and light-induced aggregation, where one led to metabolic changes [35], and the second led to the physical damage of the nervous system [11]. In *Drosophila*, transgenic animals expressed the 42-amino-acid human Aβ peptide fused to Cryptochrome 2 and the fluorescent protein mCherry (Aβ^1−42^-CRY2-mCh, Figure 1a). Expression in *Drosophila* used the GAL4/UAS system [37] with Elav-Gal4 driving expression in neurons (Appendix A, Figure 1). Glial cells were imaged using the QUAS system with repo-QF2 driving QUAS-GFP [40]. We imaged embryonic neurogenesis in detail, and observed the physical breakdown of the central nervous system upon Aβ oligomerization (Appendix A, Figure 1b,c). In lifespan analysis, we observed a rescue of Aβ-induced lifespan shortening through treatment with lithium but were unable to introduce lithium into embryos to test its effect on neurogenesis [11]. We proposed that the rescue may be due to Wnt signaling and attempted to test this in our embryonic system by expressing Wingless (Wg or Wnt1) in developing embryos. However, the expression of Wg in the developing nervous system was not successful, as embryos ceased to develop (Appendix A, Figure 1d,e). In order to establish the effect of Wnt signaling on Aβ-activity, we moved to an adult system where Wnt is involved in tissue homeostasis through its ability to maintain stem cells, the *Drosophila* gut.

The *Drosophila* gut has rapidly been established as a simple, accessible model for homeostasis, regeneration, and proliferation [60]. The tissue is composed of four cell types: enterocytes (ECs or absorptive cells), enteroendocrine (EEs or secretory cells), enteroblasts (EBs or transit amplifying cells) and intestinal stem cells (ISCs). ISCs rest on the external surface of the gut epithelium away from the gut lumen, and divide symmetrically to generate more ISCs, or asymmetrically to form EBs (Figure 2a) [61]. We confirmed the presence of gut Wg expression by imaging the midgut where a wg-Gal4 enhancer drove the expression of GFP (Figure 2b). As previously shown, expression was most prevalent in the compartment boundaries and stem cell niches [20,21]. In normal guts, ISCs are sparse, and are specifically marked by the expression of *escargot*. Therefore *esg* can be used to drive exogenous GAL4 and activate UAS-GFP or another UAS driven gene specifically in ISCs [45]. We expressed Aβ^1−42^-CRY2-mCh in this tissue and imaged the effect on ISCs. In flies kept in the dark where optogenetic oligomerization was inactive, there was a dramatic increase in GFP-positive cells, especially in transit amplifying, EBs (Figure 3a–a”, wildtype guts show an average of five ISCs and no EBs per imaged segment, whereas Aβ^1−42^-CRY2-mCh (Figure 3b–b”) expressing guts show ~9 ISCs and ~24 EBs per imaged segment). This effect was completely abrogated by the co-expression of Wg along with Aβ^1−42^-CRY2-mCh, where we observed an increase in total ISCs, but no increase in EBs (Figure 3c–c”, Aβ^1−42^-CRY2-mCh + Wg expressing guts showed ~14 ISCs and ~2 EBs). We compared the effect on flies kept in the dark (no optogenetic oligomerization) to flies exposed to light (blue light-induced CRY2-based oligomerization). We observed a small increase in GFP-positive EBs (Figure 3d–d”, Aβ^1−42^-CRY2-mCh + Light expressing guts showed on average 5 ISCs and 34 EBs) and a clear loss of EBs, as well as an increase in the number of ISCs following co-expression of Wg along with Aβ^1−42^-CRY2-mCh (Figure 3e–e”, Aβ^1−42^-CRY2-mCh + Wg + Light expressing guts showed on average 22 ISCs and 2 EBs). The expression of Wg alone led to an increase in ISC numbers (Figure 3f–f”, Wg expressing guts showed on average 15 ISCs and 2 EBs). Together, these findings suggest that either the overexpression or light-induced oligomerization of Aβ^1−42^-CRY2-mCh led to asymmetric stem cell divisions and dramatically altered tissue homeostasis. However, most importantly, tissue homeostasis was restored following the overexpression of Wg.

The expression of Aβ^1−42^-CRY2-mCh in whole organisms, or specifically in the nervous system, led to decreased life and health spans in a light-dependent manner [11]. We postulated that the loss of homeostasis observed in the *Drosophila* gut would be detrimental to adult survival, and so tested the lifespans of flies expressing amyloid and Wg in ISCs. We observed a significant decrease in the survival of flies expressing Aβ^1−42^-CRY2-mCh grown in the light (Figure 4). This effect was rescued by the expression of Wg alongside. Wg expression alone was detrimental as well, but the effect was equivalent to Aβ^1−42^-CRY2-mCh combined with Wg, suggesting that Wg rescued the Aβ^1−42^-CRY2-mCh effects, but still led to other detriments. Please note the increase in ISCs observed in Wg overexpressing guts suggesting ISC proliferation, perhaps leading to stem cell exhaustion, which may explain the reduction in lifespan for flies overexpressing Wg (Figure 3).

The finding that Wg rescued both the homeostatic and lifespan deficits caused by Aβ expression in ISCs strongly supports our hypothesis that the inhibition of glycogen synthase kinase 3 (GSK-3) by lithium, leading to the subsequent activation of the Wnt pathway, was responsible for the observed differences in survival (Figure 4c) [11]. To identify the potential pathways and mechanisms responsible for this rescue, we collected *Drosophila* guts from flies exposed to light, expressing Aβ^1−42^-CRY2-mCh (Aβ), Wg alone (Wg), Aβ^1−42^-CRY2-mCh and Wg together (WgAβ), or control flies (WT) expressing only a fluorescent protein, and performed RNA-seq (Figure 5a). Our analysis identified 2800 genes (FDR < 10%) as significantly differentially expressed across the conditions (Figure 5b, Appendix A). The clustering of these genes identified six distinct clusters, each representing groups of genes with similar expression profiles across the four conditions investigated (Figure 5b,c).

Each of the six clusters was enriched for distinct pathways and processes (Figure 5d,e, Appendix A). Cluster I (N = 440) contained genes that were upregulated by expression of Aβ^1−42^-CRY2-mCh, but were repressed by increased Wnt signaling. Genes in this cluster were associated with peroxisome, transport, and metabolic processes (Appendix A). Multiple studies have previously directly implicated the peroxisome in mediating AD pathology [62], and have implicated metabolic changes in the disease [63]. Conversely, the genes in Cluster II (N = 273) consisted of genes repressed by Aβ^1−42^-CRY2-mCh, but whose expression was rescued following the activation of Wnt signaling. This cluster was enriched for processes associated with autophagy, lysosomal activity (Appendix A), and the regulation of signaling pathways whose activation has previously been found to be protective against AD, e.g., EGFR, FoxO, and mTOR [23,64,65]. Cluster III (N = 785) and Cluster IV (N = 630) consisted of genes that were Wnt-repressed or Wnt-activated respectively, i.e., were up- or down-regulated following the overexpression of Wg, but the expression of Aβ^1−42^-CRY2-mCh had no effect on their expression. Cluster III was enriched for ribosome and translation-related processes, whereas Cluster IV was enriched for the Wnt signaling pathway and contained known Wnt target genes (e.g., *Notum*). Cluster V (N = 285) represented genes that were downregulated in the WT condition but were upregulated in all other conditions, while the genes in Cluster VI (N = 387) displayed the opposite pattern. It is likely that these two clusters simply represent the effects of perturbing the system. Clusters I and II represent groups of genes whose expression is affected by the expression of Aβ and is either repressed or rescued by the activation of Wnt signaling, and likely contain genes responsible for the observed differences in lifespan.

Within Cluster I, we identified several potential targets (Figure 5f), including CG5214, a succinyltransferase that regulates post-translational modifications of proteins and is associated with aging [66,67], and the transcriptional coactivator spargel (*srl*), the *Drosophila* homolog of *PGC-1αβ*–a gene involved in mitochondrial homeostasis and Insulin-TOR signaling, which has been implicated in AD pathogenesis [68,69]. We further observed genes related to innate immune activation, such as Persephone (*psh*), a serine protease that is involved in regulating Toll pathway activation [70]. Overall, our transcriptomic analysis showed Wnt rescuing the expression of components of key neuroprotective pathways, while acting against most of the known processes involved in AD pathogenesis.

In AD, inflammation plays an important role in disease progression [71]. The upregulation of *psh* in Aβ is indicative of the activation of the Toll pathway, a key pathway involved in inflammation and innate immunity. We used the Toll receptor, a key activator of innate immune response, to assay the inflammatory activity as it relates to lifespan. We used an optogenetic version of Toll; we had previously constructed and tested its effect on fly lifespans [46]. We expressed Toll-Cry2-mCherry in adult flies using the esg-Gal4 system to target ISCs, and in the whole fly using Arm-Gal4. Whole animal expression of Toll-Cry2-mCherry along with light exposure led to rapid death (Figure 6). The expression only in ISCs was less damaging, but the activation of the Toll pathway by Toll-Cry2-mCherry led to a reduction in lifespan, comparable to the effect of Aβ^1−42^-CRY2-mCh. The effects were comparable when tested in flies kept in the dark. Overall, modulating the level of innate immune response led to lifespan shortening.

Finally, we looked at the effect of *persephone* on *Drosophila* lifespan, as this gene was a major target identified by the RNA sequencing experiment (Figure 5f). Loss of *persephone* should lead to lower inflammation, since Toll activity would be reduced [70,72]. We observed an increase in median lifespan of *Drosophila* expressing RNAi to knockdown *psh* as compared to control flies (Figure 7). In addition, we looked at a stronger loss of immune response by knocking down the *Dorsal-related immunity factor* (*Dif*) [73]. Loss of *Dif* led to a reduction in lifespan, suggesting that a strong loss of immune response remains detrimental (Figure 7). Overall, we find that a reduction in inflammation is beneficial, whereas strong inflammation or strong loss of immune response remains detrimental.

## 4. Discussion

In an update to our previously published study, we looked at the mechanism that lithium used to extend the lifespan of *Drosophila* expressing Aβ^1−42^-CRY2-mCh. We postulated that the Wnt pathway was the key component to this based on previous studies [22,23,34,74]. However, we were unable to test the model in embryonic neurogenesis as the application of lithium and Wnt to embryos proved either technically or genetically problematic, and instead we investigated this in the *Drosophila* gut, an adult homeostasis model. The overexpression of Wnt in ISCs was found to ameliorate the homeostatic and lethal consequences of Aβ^1−42^-CRY2-mCh expression. We found several metabolic, proteostatic and inflammatory pathways to be activated by Aβ^1−42^-CRY2-mCh and repressed by Wnt. We propose that Wnt’s role in promoting homeostasis in stem cells can be extended to prevent some of the detrimental effects of Aβ^1−42^-CRY2-mCh expression by preventing Toll pathway activation. We show that Toll activation is just as detrimental as Aβ^1−42^-CRY2-mCh expression, while *psh* knockdown is beneficial, and propose that Wnt prevents Toll hyperactivation, leading to amelioration of amyloid-based detriments.

As we and others have previously proposed that lithium could be used as a treatment for AD [75,76], by narrowing the effect of lithium to Wnt activation, this creates the possibility of using Wnt activating drugs in AD treatment, especially as one has recently been shown to be life extending [77]. Most importantly, the homeostatic role of Wnt is not the only developmental signal that promotes homeostasis and prevents amyloid detriments. Recently, a role for Hedgehog was discovered in glial cells to promote lifespan and prevent amyloid-dependent neurodegeneration [78]. These findings together suggest that stem cell homeostasis and the re-activation of developmental pathways may be key mechanisms that could be targeted to prevent neurodegeneration.

## Figures and Tables

**Figure 1 cells-11-00281-f001:**
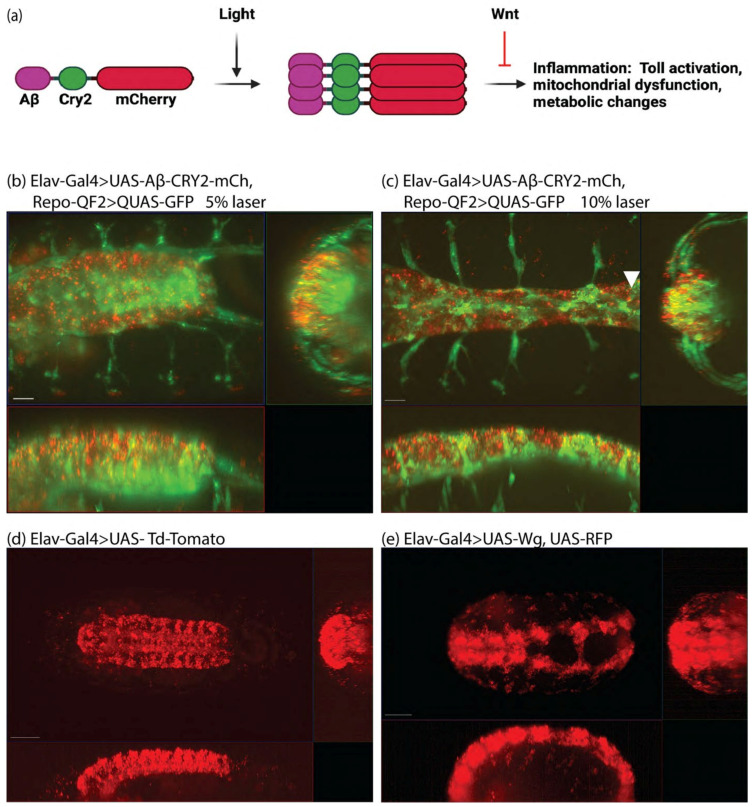
The optogenetic amyloid system in embryos. (**a**) Scheme of light-induced Aβ-Cry2mCh clustering in *Drosophila melanogaster* resulting in activation of inflammatory processes which can be rescued by Wnt. Stills from Appendix A (**b**), Appendix A (**c**), Appendix A (**d**), Appendix A (**e**). Arrowhead is pointing to nervous system constriction.

**Figure 2 cells-11-00281-f002:**
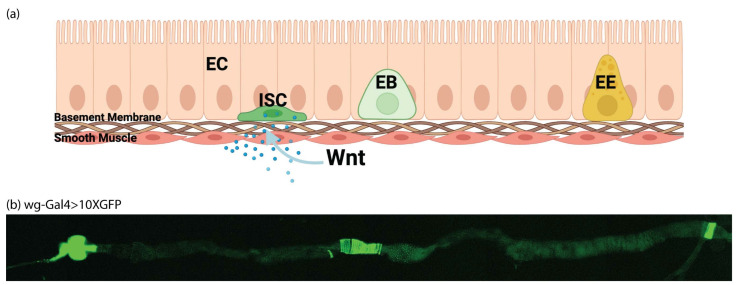
Wnt expression in the *Drosophila* gut. (**a**) Model of the *Drosophila* gut comprising enterocytes (ECs), enteroendocrine (EEs), enteroblasts (EBs) and intestinal stem cells (ISCs). ISCs rest on the external surface of the gut epithelium away from the gut lumen and divide symmetrically to make more ISCs or asymmetrically to form EBs. (**b**) Representative midgut section of a wg-Gal4 > 10× GFP fly.

**Figure 3 cells-11-00281-f003:**
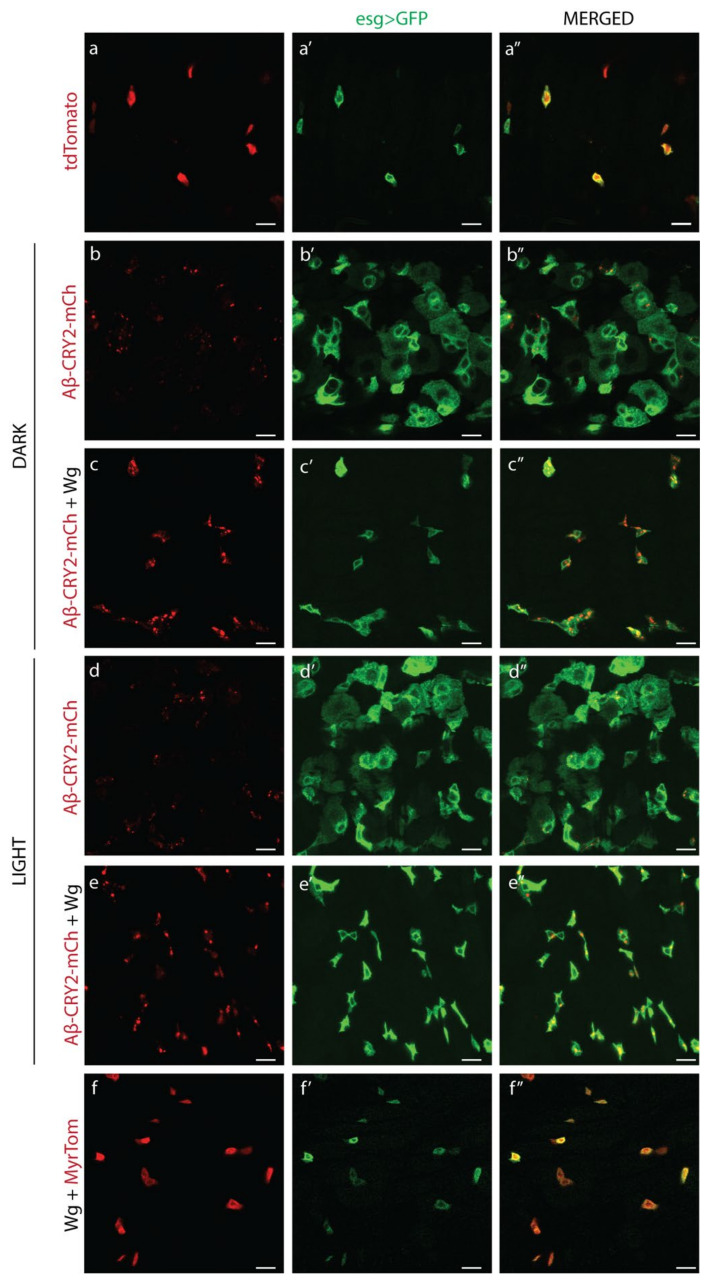
Wnt and amyloid expression in ISCs. Images of midgut sections of (**a**–**a”**) esg-Gal4 > UAS-GFP, UAS-Td-Tomato flies, (**b**–**b”**) esg-Gal4 > UAS-GFP, UAS-Aβ^1−42^-CRY2-mCh flies kept in the dark, (**c**–**c”**) esgGal4 > UAS-GFP, UAS-wg, UAS-Aβ^1−42^-CRY2-mCh flies kept in the dark, (**d**–**d”**) esg-Gal4 > UAS-GFP, UAS-Aβ^1−42^-CRY2-mCh flies kept in the light, (**e**–**e”**) esg-Gal4 > UAS-GFP, UAS-wg, UAS-Aβ^1−42^-CRY2-mCh flies kept in the light and (**f**–**f”**) esg-Gal4 > UAS-GFP, UAS-wg, UAS-myr-Tomato flies. Scale bar represents 10 µm.

**Figure 4 cells-11-00281-f004:**
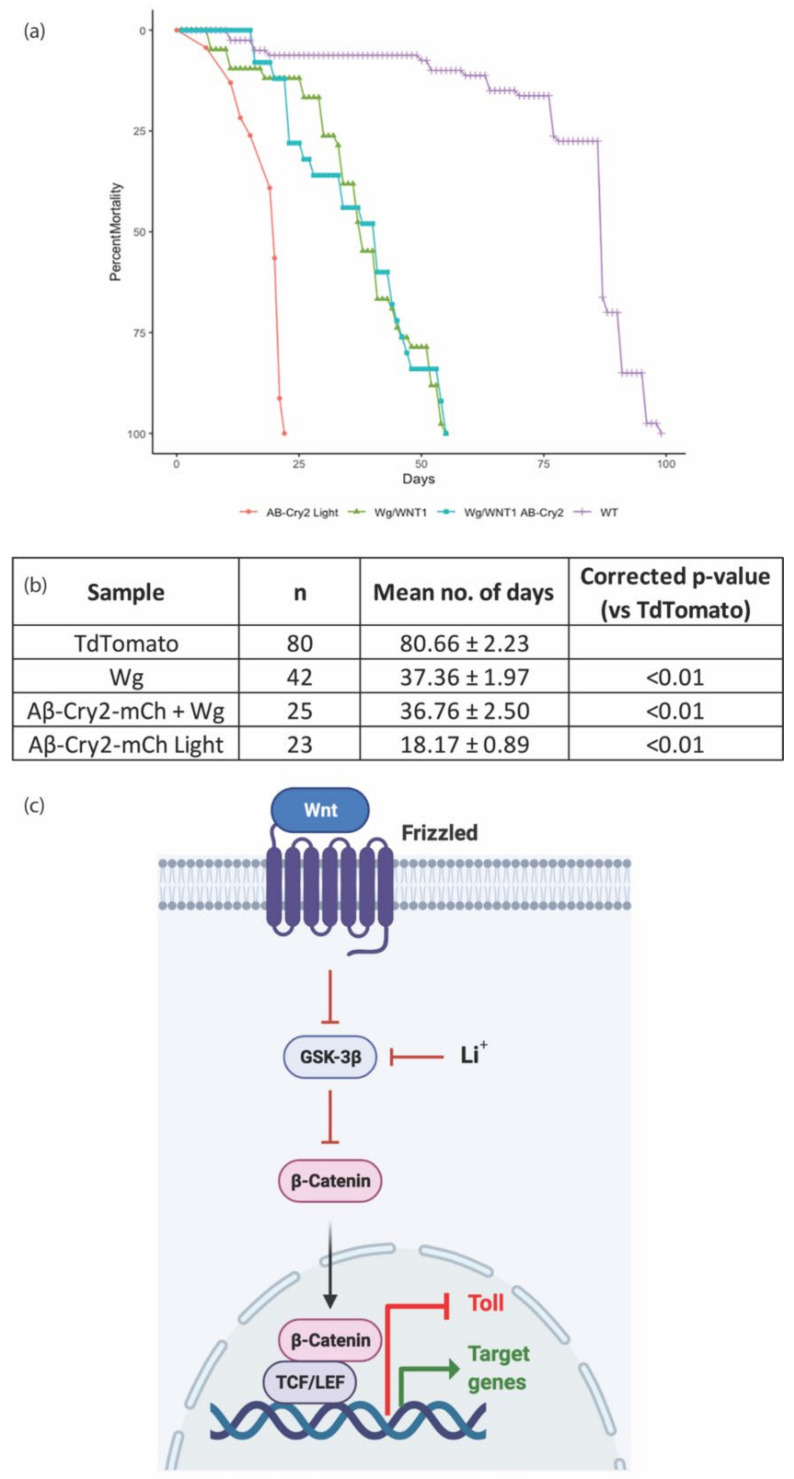
Lifespan decrease rescued by Wnt expression. Lifespan analysis of flies expressing TdTomato (control), Wg, Aβ-CRY2-mCh + Wg and Aβ-CRY2-mCh in gut stem cells. Genotype: *TdTomato*—esg-Gal4>UAS-GFP, UAS-Td-Tomato; *Wg*—esg-Gal4> UAS-GFP, UAS-wg, *Aβ-CRY2-mCh + Wg*—esg-Gal4> UAS-GFP, UAS-wg, UAS-Aβ^1-42^-CRY2-mCh; *Aβ-CRY2-mCh*—esg-Gal4> UAS-GFP, UAS- Aβ^1-42^-CRY2-mCh. (**a**) The lifespan curves for the various genotypes, and (**b**) table showing the number of subjects with mean lifespans and *p*-values. (**c**) Diagram showing lithium-induced activation of the Wnt pathway inhibiting Toll signalling. Raw data provided in Appendix A.

**Figure 5 cells-11-00281-f005:**
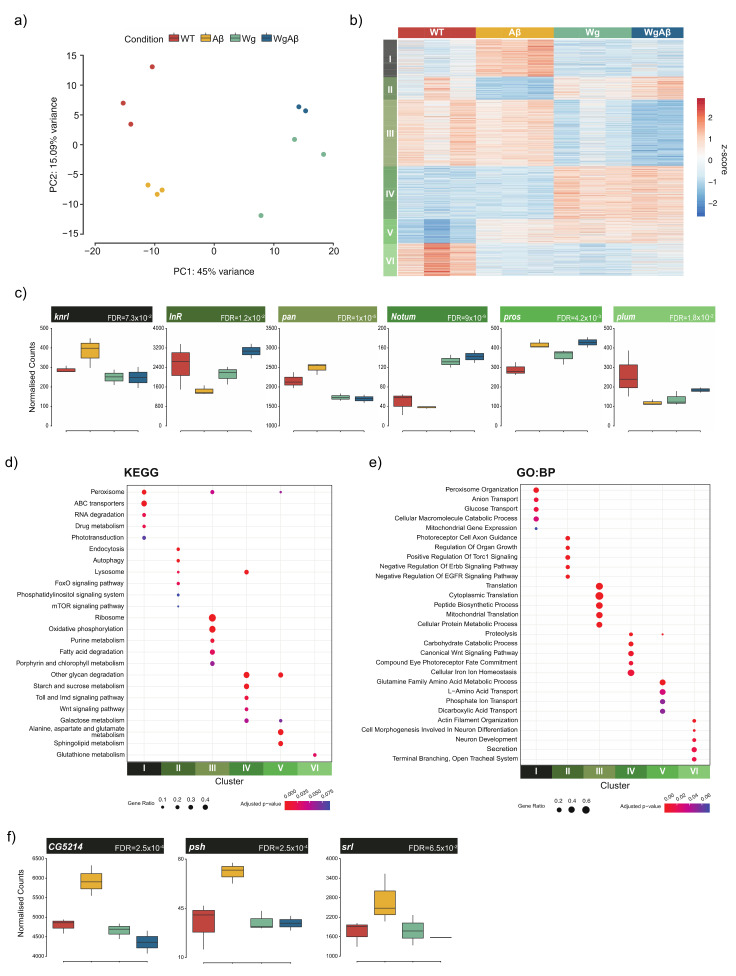
Gene expression changes in response to amyloid and Wnt. (**a**) Principal components analysis of RNA-seq data reveals clear separation of samples by experimental condition (**b**) Clustering of differentially expressed genes identifies six clusters with distinct expression patterns across the four conditions (**c**) Expression profiles of representative genes for each of the six clusters (**d**) KEGG pathway enrichments identifies key developmental and homeostatic pathways associated with the individual clusters (FDR < 10%) (**e**) Biological process (GO:BP) analysis identifies distinct processes enriched in each of the clusters (FDR < 10%). (**f**) Expression profiles of genes from Cluster I which are relevant to AD biology.

**Figure 6 cells-11-00281-f006:**
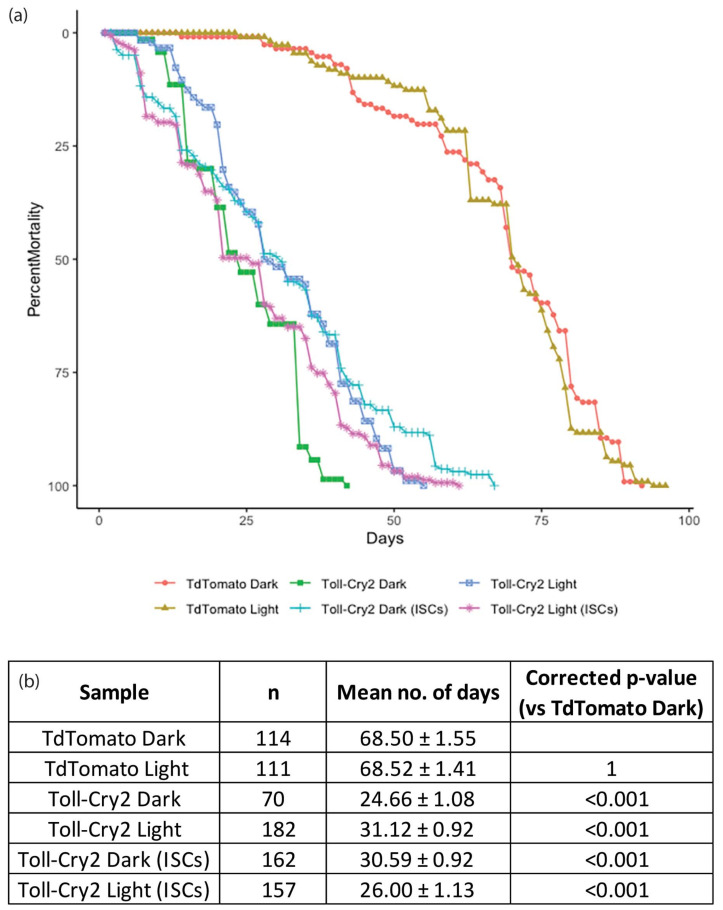
Activated Toll reduces lifespan. Lifespan analysis of TdTomato (control) dark and light, Toll-Cry2-mCh dark and light in the whole fly, Toll-Cry2-mCh dark and light expression in the ISCs only. Genotype: TdTomato—Arm-Gal4 > UAS-Td-Tomato; Toll-Cry2—ArmGal4 > UAS-Toll-Cry2-mCh; Toll-Cry2 (ISCs)—esg-Gal4 > UAS-GFP, UAS-Toll-Cry2-mCh. (**a**) The lifespan curves for the various genotypes, and (**b**) table showing the number of subjects with mean lifespans and *p*-values. Raw data provided in Appendix A.

**Figure 7 cells-11-00281-f007:**
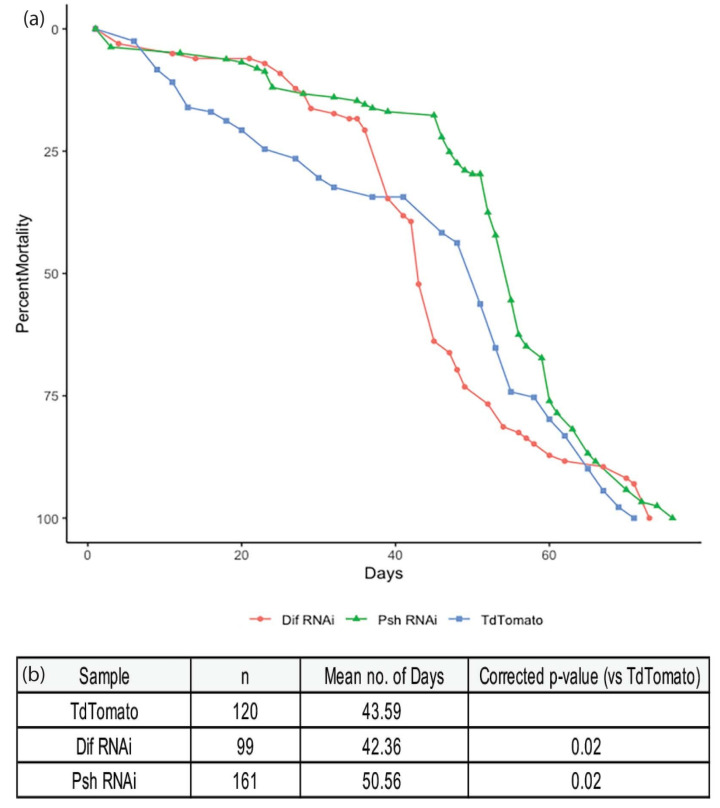
Knockdown of persephone extends lifespan. Lifespan analysis of TdTomato (control), psh-RNAi and Dif-RNAi in the whole fly. Genotype: TdTomato—Arm-Gal4 > UAS-Td-Tomato; Dif RNAi—ArmGal4 > UAS-Dif RNAi; Psh RNAi—Arm-Gal4 > UAS-Psh RNAi. (**a**) The lifespan curves for the various genotypes, and (**b**) table showing the number of subjects with mean lifespans and *p*-values. Raw data provided in Appendix A.

## Data Availability

RNA-seq data generated from this study have been deposited to NCBI GEO (GSE181844) https://www.ncbi.nlm.nih.gov/geo/query/acc.cgi?acc=GSE181844 (accessed on 10 August 2021). All code necessary to recreate the results from the RNA-seq analysis is available from: https://github.com/harmstonlab/Ab (accessed on 5 August 2021).

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
