# Peer review of "Wnt Signaling Rescues Amyloid Beta-Induced Gut Stem Cell Loss"

_cells, 2022, doi:10.3390/cells11020281_

Round 1

Reviewer 1 Report

This is nice work with sufficient evidence and logical writing. Dr. N Tolwinski's lab focuses on neurogenesis and metabolism of AD and this study further proved that Lithium treatment contributes to Wnt activation and inhibition of inflammation in Drosophila. All data are clear to follow and reasonable. I believe this MS could be published in the present form.

Author Response

Reviewer 1:

This is nice work with sufficient evidence and logical writing. Dr. N Tolwinski's lab focuses on neurogenesis and metabolism of AD and this study further proved that Lithium treatment contributes to Wnt activation and inhibition of inflammation in Drosophila. All data are clear to follow and reasonable. I believe this MS could be published in the present form.

We thank the reviewer for the kind words. 

Reviewer 2 Report

The manuscript by Kaur et al investigated the role of Wnt and Toll signaling in homeostasis and inflammation related to Aβ detrimental effects by using Drosophila gut stem cells. It is very challenging and significant study to explore novel AD treatment. However, it seems that the verification in this article is insufficient and illogical. Furthermore, the authors didn’t describe the trial number of independent experiments in some FIgures and didn’t also described appropriate statistical analysis in materials and methods section. Results in this article are unreliable because of above reasons. Therefore, their study lacks reliability and does not warrant publication in cells. Below are some of the major concerns.

Major comments:

  1. Overall, explanations of results were insufficient in results part, especially Figure 1b and c. Moreover, reviewer couldn’t find the description of Figure 5e and 7. And some images didn’t include scale bar.
  2. In this article, the author proposed that the rescue may be due to Wnt signaling and attempted to test this in our embryonic system by expressing Wingless in developing embryos. However, why authors focused on Wnt signaling was not logically explained.
  3. The author should more explain whether gut stem cells and neural stem cell are correlated in AD. Reviewer couldn't understand the necessity of using gut stem cell. If author claims that Wnt signaling restore Aβ detrimental effects in brain, another approach should be performed for instance mouse model or culture system.

Author Response

Reviewer 2:

The manuscript by Kaur et al investigated the role of Wnt and Toll signaling in homeostasis and inflammation related to Aβ detrimental effects by using Drosophila gut stem cells. It is very challenging and significant study to explore novel AD treatment. However, it seems that the verification in this article is insufficient and illogical. Furthermore, the authors didn’t describe the trial number of independent experiments in some FIgures and didn’t also described appropriate statistical analysis in materials and methods section. Results in this article are unreliable because of above reasons. Therefore, their study lacks reliability and does not warrant publication in cells. Below are some of the major concerns.

 Thank you for the comments and concerns.  We have attempted to address these issues by providing further explanations and the raw data as a supplement.  All experiments were based on a minimum of triplicate.  We apologize if we failed to make this obvious in the original manuscript.

Major comments:

  1. Overall, explanations of results were insufficient in results part, especially Figure 1b and c. Moreover, reviewer couldn’t find the description of Figure 5e and 7. And some images didn’t include scale bar.

For figure 1, these are stills from the videos uploaded in the supplement.  These did not upload correctly and we apologize for the inconvenience.  We have added the descriptions and scale bars.

  1. In this article, the author proposed that the rescue may be due to Wnt signaling and attempted to test this in our embryonic system by expressing Wingless in developing embryos. However, why authors focused on Wnt signaling was not logically explained.

I am not sure what the reviewer is referring to here.  We wanted to test what mechanism is downstream of Lithium.  As a GSK3 kinase inhibitor, Wnt is a possible target.  This is why we looked at Wnt signaling activation.  We have tried to make this clearer in the text.

  1. The author should more explain whether gut stem cells and neural stem cell are correlated in AD. Reviewer couldn't understand the necessity of using gut stem cell. If author claims that Wnt signaling restore Aβ detrimental effects in brain, another approach should be performed for instance mouse model or culture system.

I am certain that a mouse model would be a great approach, but that experiment is well outside the scope of this Drosophila study.  We explain in the manuscript that the approach is to test new models with simple imaging properties. 

Reviewer 3 Report

The article Wnt signaling rescues amyloid beta induced stem cell loss presents the importance of Wnt signaling pharmacological activation for stem cells promotion. Data are interesting and promising in the view of possible therapeutic approach concerning Alzheimer's Disease (AD). However, several issues should be corrected/discussed.

Major issues:

  1. Currently the text in not so easy to follow. Some suggestions:
    a) in Introduction I encourage to add paragraph about current knowledge of Wnt signaling role in AD; authors only shortly mentioned their experience;
    b) in Materials and Methods Figures 4 and 6 are located - its location should be classically organised in Results section;
    c) Captions for Figure 4 and 6 should be more informative and contain description for (a), (b), (c)
    d) Figures S1 and S2 should be included in Supplementary Materials

2. In Figure 4 lifespan decrease is shown. Mean for Aβ is the lowest (18) and for control the highest (81). Wg expression partly rescue the negative effects of Aβ (37), but expression of Wg also reduced the lifespan to 37 days. It suggest negative effects of Wnt signaling overexpression, characteristic for many kinds of cancer, including CNS tumors. Please discuss this results.

Minor issues:

  1. Figure 4c - please change Li+ --> Li+
  2. Figure 1 caption - line 160: Schematic of --> Scheme of
  3. Figure 3 Scale bar is missing (except a'')
  4. Reference to Figure 7 in main text is missing
  5. Additional spaces in the manuscript should be deleted.
  6. I do not see supplementary tables in the review system (I asked editor to check it) and I have problem to play the video.

Author Response

Reviewer 3:

The article Wnt signaling rescues amyloid beta induced stem cell loss presents the importance of Wnt signaling pharmacological activation for stem cells promotion. Data are interesting and promising in the view of possible therapeutic approach concerning Alzheimer's Disease (AD). However, several issues should be corrected/discussed.

We thank the referee for the positive comments.  We address the points below and in the text.

Major issues:

  1. Currently the text in not so easy to follow. Some suggestions:
    a) in Introduction I encourage to add paragraph about current knowledge of Wnt signaling role in AD; authors only shortly mentioned their experience;

We have added a paragraph on this topic to the introduction.

  1. b) in Materials and Methods Figures 4 and 6 are located - its location should be classically organised in Results section;

Thank you.  We have only made the preliminary version of the manuscript.  The page proofs will distribute the figures properly.

  1. c) Captions for Figure 4 and 6 should be more informative and contain description for (a), (b), (c)

Thank you.  We have added these to the legends.

  1. d) Figures S1 and S2 should be included in Supplementary Materials

These will be moved in the final proofs.

  1. In Figure 4 lifespan decrease is shown. Mean for Aβ is the lowest (18) and for control the highest (81). Wg expression partly rescue the negative effects of Aβ (37), but expression of Wg also reduced the lifespan to 37 days. It suggest negative effects of Wnt signaling overexpression, characteristic for many kinds of cancer, including CNS tumors. Please discuss this results.

We have added a possible explanation dealing with stem cell exhaustion.  This is the most common reason that flies live shorter lifespans in our hands.

Minor issues:

  1. Figure 4c - please change Li+ --> Li+

Done

  1. Figure 1 caption - line 160: Schematic of --> Scheme of

Done

  1. Figure 3 Scale bar is missing (except a'')

Done

  1. Reference to Figure 7 in main text is missing

Done

  1. Additional spaces in the manuscript should be deleted.

Done

  1. I do not see supplementary tables in the review system (I asked editor to check it) and I have problem to play the video.

There was an issue with the upload.  We apologize for this and have rectified it in the resubmission. 

Round 2

Reviewer 3 Report

Thank you very much for the answers and corrections you have prepared.

Minor issue:

Caption for Figure 4, please correct as following:

Figure 4. (...) UAS- Aβ1-42-CRY2-mCh. (c) Schematic of Lithium induced activation of the Wnt pathway to inhibit Toll signalling. (a) The lifespan curves for the various genotypes, and (b) table showing the number of subjects with mean lifespans and p-values. (c) Scheme showing Lithium induced activation of the Wnt pathway to inhibit Toll signalling.  Raw data provided in Supplemental Table 4.

Author Response

We have corrected the figure legend.